# Investigating the role of FOX gene family in development and stress response in *Labeo rohita*: A multi-faceted analysis of phylogeny and genome characterization

Saima Naz[1], Urwah Ishaque[1], Ahmad Manan Mustafa Chatha[2], Muhammad Farooq[3], Qudrat Ullah[4], Shabana Naz[5]*, Marco Ragni[3], Ibrahim Alhidary[6]

**1** Department of Zoology, The Government Sadiq College Women University, Bahawalpur, Pakistan, **2** Department of Entomology, Faculty of Agriculture and Environment, The Islamia University of Bahawalpur, Punjab, Pakistan, **3** Department of Soil, Plant and Food science, University of Bari, Aldo Moro, Italy, **4** Department of Theriogenology, Cholistan University of Veterinary and Animal Sciences, Bahawalpur, Pakistan, **5** Department of Zoology, Government College University Faisalabad, Pakistan, **6** Department of Animal Produciton, College of Food and Agriculture Science, King Saud University, Riyadh, Saudi Arabia

* drshabananaz@gcuf.edu.pk

## Abstract

The forkhead box (FOX) gene family of transcription factors regulates muscle development, immune responses, and metabolic processes across species. Despite extensive studies on FOX genes in other organisms, their evolutionary and functional roles in *Labeo rohita*, an economically and ecologically important freshwater fish, remain unclear. Owing to its unique physiological and ecological traits, *L. rohita* is an ideal model for exploring these roles. Here, we present the first computational analysis of the FOX gene family in *L. rohita*, identifying 21 FOX genes. Physicochemical analysis revealed that most FOX proteins have a basic nature except for FOX A3, D3, I2, O1, O3, O4, P1, and P2. Instability index analysis indicated that all FOX proteins are unstable (values > 40), while hydrophobicity assessment showed that except FOX O1, all proteins are hydrophobic. Phylogenetic analysis grouped FOX homologs into 11 major clades with other vertebrates. All proteins exhibited structural homogeneity by sharing the Forkhead Box domain. Gene structure comparisons revealed seven duplicated pairs, and Circos analysis demonstrated organization into 20 clusters. This study highlights the critical roles of FOX genes and fills a significant knowledge gap, providing a foundation for future functional and phylogenomic studies with implications for aquaculture and evolutionary biology.

## 1. Introduction

The Forkhead box (FOX) gene family is known for its versatility and evolutionary conservation among transcription factors, distinguished by a unique forkhead or

**Data availability statement:** The data are all contained within the manuscript.

**Funding:** The author(s) received no specific funding for this work.

**Competing interests:** The authors have declared that no competing interests exist.

winged-helix domain [1]. Since its identification in *Drosophila melanogaster* in the early 1990s, interest in FOX genes has surged, highlighting their essential regulatory functions across various species, including invertebrates, vertebrates, and humans [2]. These genes are crucial for regulating gene expression related to key biological processes such as cell proliferation, differentiation, apoptosis, and metabolism [3]. In metazoans, the FOX family plays a significant role in early embryonic development, tissue specification, and organogenesis, underscoring its importance in both vertebrate and invertebrate systems [4].

FOX genes are characterized by their distinct helix-turn-helix DNA binding domain, enabling them to attach to specific DNA sequences and regulate gene networks crucial for development and maintaining homeostasis [5]. The FOX family is categorized into subclasses labeled with letters (A to S), determined by sequence similarities found within the forkhead domain [6]. In the last twenty years, significant research has concentrated on the evolutionary divergence of FOX genes and their functions in developmental biology. For instance, investigations in humans and model organisms like mice and zebrafish have shown that FOX genes play crucial roles in numerous processes, including neural development, organogenesis, and the regulation of immune responses. These results indicate that the FOX gene family acts as a master regulator of various developmental and physiological pathways [7].

In zebrafish (*Danio rerio*), a key model organism for investigating vertebrate development, the FOX family includes 64 isoforms spanning several subfamilies, such as FOX A, FOX B, FOX C, FOX D, and FOX P. Research involving zebrafish has offered valuable insights into the varied functions of these genes during early development and neurogenesis. For instance, FOX D3 is crucial for the differentiation of neural crest cells, while FOX P2 is vital for speech and language development in humans and the regulation of neural circuits in vertebrates [8]. The role of FOX genes in stress response, especially concerning oxidative stress and environmental adaptations, has garnered increased interest in recent years. For example, Zhang et al. (2023) performed a thorough analysis of the Corsac FOX (*Vulpes corsac*) genome, uncovering genes linked to stress response mechanisms [9]. Despite these advancements, the exact molecular mechanisms and target genes regulated by FOX proteins are still not well understood, especially in non-model species like teleost fish.

*Labeo rohita*, commonly referred to as rohu, is an essential freshwater species within the Indian major carps (IMCs) group and plays a crucial role in aquaculture throughout South Asia. As a widely cultivated species, *L. rohita* significantly contributes to food security and the livelihoods of millions in the region [10]. However, similar to other fish species, *L. rohita* is vulnerable to environmental stressors like temperature fluctuations, hypoxia, and water pollution, which can negatively impact its growth, survival, and reproduction [11]. Gaining insights into the genetic and molecular foundations of stress response and adaptation in *L. rohita* is essential for creating strategies to improve its resilience and productivity in the face of changing environmental conditions. However, there is still a considerable gap in understanding the regulatory networks that govern stress responses in this species.

Considering the crucial roles of FOX genes in regulating development, stress responses, and immune functions in other vertebrates, it is vital to explore their functions in *L. rohita*. The FOX gene family in *L. rohita*, similar to other teleost species, likely plays a key role in facilitating developmental processes and responses to environmental challenges. Research in teleost fish has shown that members of the FOX gene family, such as FOX O and FOX A, are involved in pathways that regulate cellular responses to oxidative stress and promote longevity [12]. However, comprehensive computational studies of the FOX gene family in *L. rohita* have yet to be performed, creating a significant gap in our understanding of the genetic regulation of essential physiological processes in this species.

The present study focuses on the comprehensive computational analysis of the FOX gene family in *L. rohita.* By utilizing bioinformatics tools and comparative genomics methods, we aim to identify and characterize the FOX genes in *L. rohita*, shedding light on their evolutionary relationships, molecular structures, and potential regulatory functions. The main objectives of this research include: (1) identifying the complete set of FOX genes in the *L. rohita* genome, (2) conducting phylogenetic analysis to explore their evolutionary relationships with FOX genes in other vertebrates, (3) performing motif and domain analysis to uncover structural conservation and divergence, and (4) predicting the transcription factor binding sites that regulate the expression of FOX genes under various physiological conditions.

This study will also investigate the role of FOX genes in stress responses, emphasizing their involvement in pathways related to oxidative stress, immune regulation, and apoptosis. By integrating gene expression data with in-silico analyses, we aim to explore the transcriptional regulation of FOX genes in *L. rohita* under stress conditions such as hypoxia and temperature fluctuations. Understanding the molecular mechanisms through which FOX genes mediate stress responses will provide insights into how *L. rohita* adapts to environmental stressors and could inform the development of selective breeding programs designed to enhance stress resilience in aquaculture.

Moreover, the comparative genomics aspect of this study will enable us to identify both evolutionary conservation and species-specific adaptations of FOX genes in *L. rohita*. This will contribute to a broader understanding of the functional diversification of FOX genes in teleosts and their role in vertebrate evolution. The findings from this research will not only enhance our knowledge of the FOX gene family in *L. rohita* but will also establish a framework for future studies on the molecular mechanisms governing development and stress responses in other economically significant fish species.

This comprehensive computational analysis of the FOX gene family in *L. rohita* aims to illuminate their evolutionary history, molecular functions, and regulatory roles. By investigating the FOX genes in this species, we seek to fill the knowledge gap regarding the genetic factors influencing development, stress response, and immune function in *L. rohita*. The results of this research have the potential to guide strategies for improving the growth, health, and resilience of *L. rohita* in aquaculture, contributing to more sustainable and efficient fish farming practices in South Asia. Furthermore, computational analysis of the FOX gene family in *L. rohita* offers a valuable opportunity to explore the genetic basis of development and environmental adaptation in this important aquaculture species. By elucidating the molecular mechanisms through which FOX genes regulate key physiological processes, this study will deepen our understanding of the developmental biology and stress physiology of *L. rohita* and other teleost fish. The insights gained from this research will have broader implications for vertebrate genomics and the application of molecular genetics in aquaculture and fisheries management.

## 2. Results

### 2.1. Analysis of the Phylogenetic Relationships in the FOX Gene Superfamily

To explore the evolutionary history of the FOX gene superfamily, a molecular phylogenetic analysis was performed. In total, 94 amino acid sequences from *Labeo rohita* (Rohu), *Oreochromis niloticus* (Nile tilapia), *Ctenopharyngodon idella* (Grass carp), *Danio rerio* (Zebrafish), and *Homo sapiens* (Human) were analyzed. These sequences were organized into 11 major clades: FOX H, FOX M, FOX P, FOX O, FOX F, FOX L, FOX G, FOX D, FOX C, FOX I, and FOX A, based on the percentage of homologous gene sequences (Figure 1). The results indicated that the FOX gene superfamily in

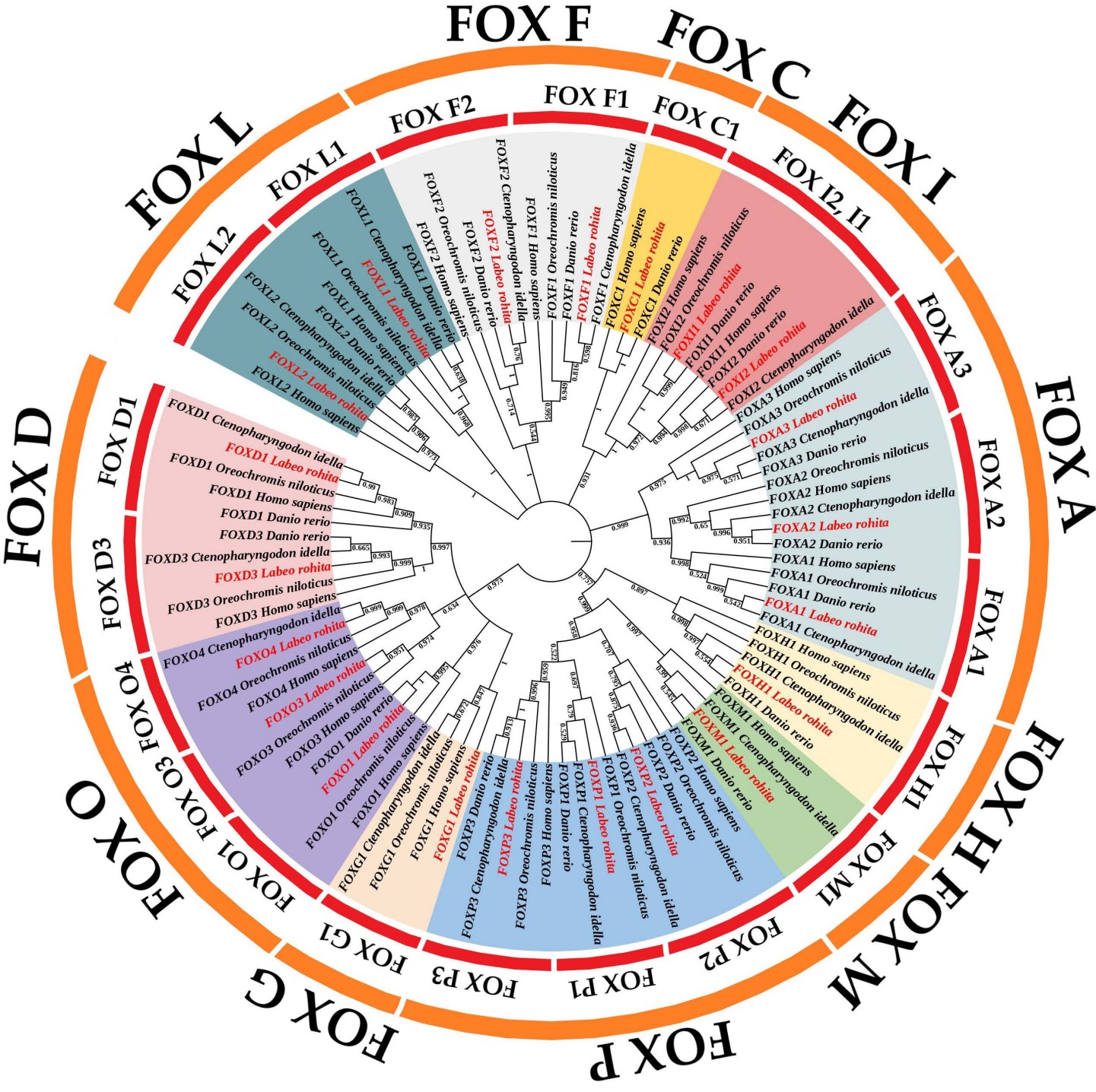

**Fig 1. Phylogenetic tree of the FOX gene superfamily based on 94 amino acid sequences from five species (*L. rohita*, *O. niloticus*, *C. idella*, *D. rerio*, and *H. sapiens*), clustering into 11 clades.**

*L. rohita* showed a higher degree of sequence similarity with *D. rerio* and *C. idella* compared to other vertebrate species (Figure 1).

## 2.2. Characterization of the Structure of FOX Genes Superfamily in *L. rohita*

To investigate the structural characteristics of the FOX gene superfamily in *L. rohita*, we conducted an analysis of the evolutionary relationships within the FOX gene superfamily, as well as an examination of motifs, conserved regions, and gene structures (Figure 2). We identified ten conserved motifs in the FOX genes of *L. rohita* through MEME analysis, with MEME-6 containing the highest number of amino acids (50). This motif was classified as the Forkhead Box protein domain using InterPro (Figure 2B; Table 1). Our findings were further supported by a comparison with the NCBI CDD database (Figure 2C). Additionally, the Forkhead Box protein domain was present in all homologs of the FOX gene super-family. The gene structure analysis revealed significant variations in the arrangement of exons, introns, and both down-stream and upstream untranslated regions (UTRs) (Figure 2D).

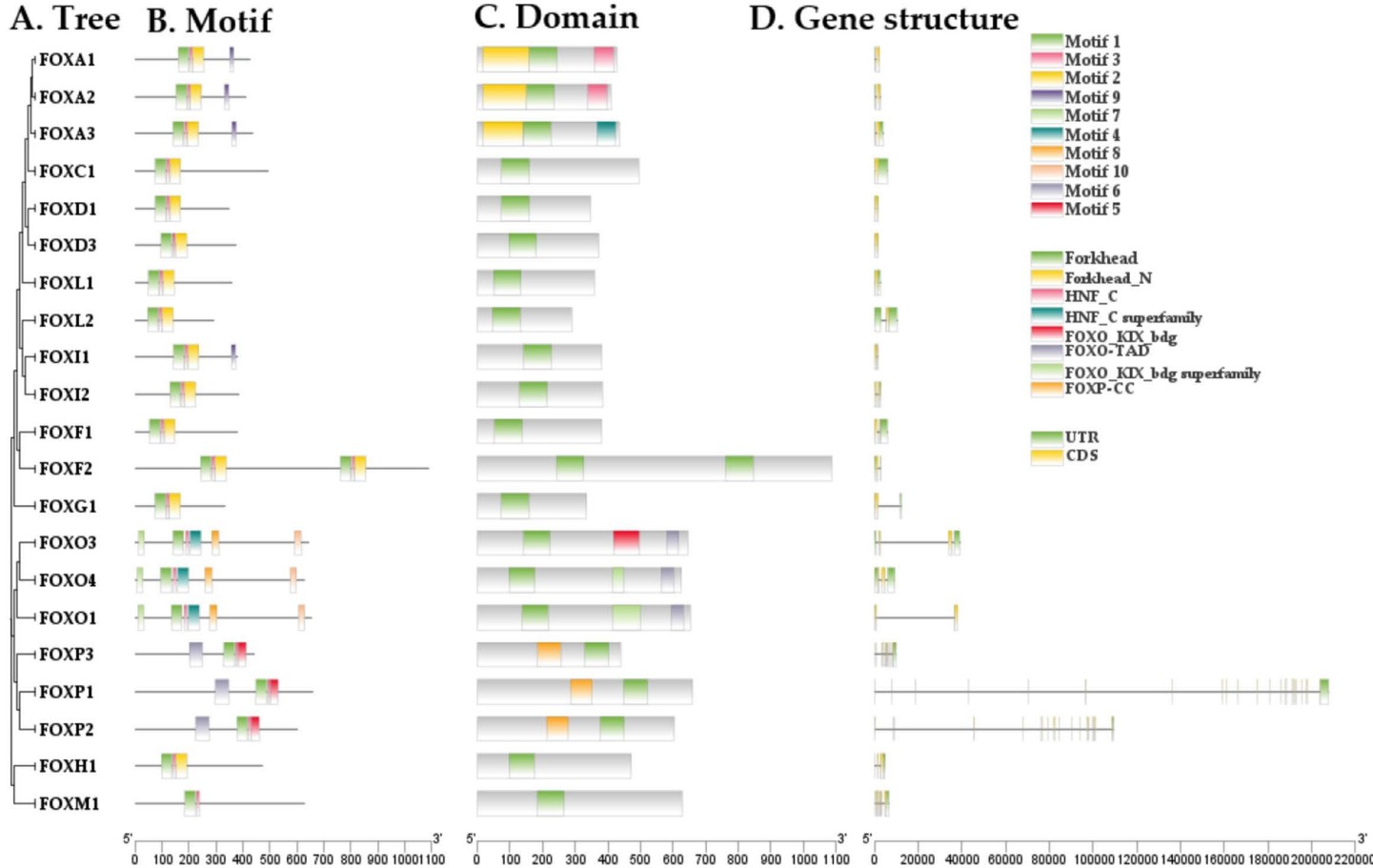

**Fig 2. Structural Analysis of the FOX gene superfamily: (A) Phylogenetic tree, (B) Motifs, (C) Domains, and (D) Gene structure in *L. rohita*.**

**Table 1. Ten highly conserved motifs identified within the FOX gene family of *L. rohita*.**

| MEME Motif | Amino Acid Sequence | Width | InterPro Domain |
|---|---|---|---|
| 1 | PYSYIALITMAIQNSPEKRLTLSEIYQWIMDRFPYYRDNK | 40 | Forkhead Box Protein |
| 2 | DCFVKVPREPGKPGKGNYWTLDPNSEBMFENGSFRRRRKRF | 41 | Forkhead Box Protein |
| 3 | WQNSIRHNLSL | 11 | Forkhead Box Protein |
| 4 | RVQNEGTGKSSWWMLNPEGGKSGKAPRRRAASMDNSSKLJK | 41 | Forkhead Box Protein |
| 5 | HKCFVRVENVKGAVWTVDELEFQKRRPQKI | 30 | Forkhead Box Protein |
| 6 | VFEDFGQFLKHLNNEHALDDRSTAQCRVQMQVVQQLEJQLSKERERLQAM | 50 | Forkhead Box Protein |
| 7 | IDPDFEPQSRPRSCTWPLPRPE | 22 | – |
| 8 | DDFDAWTAFRPRTSSNASTLSGRLSPI | 27 | – |
| 9 | DPHYSFNHPFSINNLM | 16 | – |
| 10 | FDCDVEAIJHBDLMDGEGLDFNFD | 24 | Forkhead Box Protein |

## 2.3. Physicochemical Analysis of FOX Genes in *L. rohita*

The physicochemical properties of the FOX gene family in *L. rohita* were assessed, including chromosome number, amino acid count, exon number for each peptide, isoelectric point (pI), molecular weight (MW), instability index (II), grand average of hydropathicity index (GRAVY), and aliphatic index (AI), as detailed in Table 2. The molecular weights of the FOX proteins in *L. rohita* ranged from 32705.96 to 119782.93 Da, while the pI values varied from 4.81 to 9.50. The aliphatic index values exceeded 41.83, indicating that all FOX genes in *L. rohita* possess thermostable characteristics. Most proteins exhibited basic properties, except for FOX A3, FOX D3, FOX I2, FOX O1, FOX O3, FOX O4, FOX P1, and FOX P2.

**Table 2. Physicochemical properties of the FOX gene superfamily in *L. rohita*.**

| Gene name | Chr. No | Exon count | Molecular Weight (D) | Amino acid | Isoelectric point | Instability index | Aliphatic index | GRAVY |
|---|---|---|---|---|---|---|---|---|
| FOXA1 | 17 | 2 | 46614.79 | 428 | 9.09 | 65.14 | 46.36 | 0.691 |
| FOXA2 | 17 | 2 | 45057.48 | 409 | 8.8 | 67.78 | 41.83 | 0.711 |
| FOXA3 | 18 | 2 | 47543.6 | 435 | 6.99 | 69.94 | 55.2 | 0.695 |
| FOXC1 | 2 | 1 | 54597.25 | 495 | 8.97 | 61.54 | 51.07 | 0.842 |
| FOXD1 | 5 | 1 | 37852.56 | 347 | 7.23 | 68.89 | 70.61 | 0.452 |
| FOXD3 | 6 | 1 | 40412.44 | 327 | 5.28 | 64.36 | 73.47 | 0.409 |
| FOXF1 | 18 | 2 | 41556.67 | 381 | 9.16 | 65.46 | 50.47 | 0.604 |
| FOXF2 | 2 | 2 | 119782.93 | 1087 | 9.33 | 60.16 | 64.29 | 0.43 |
| FOXG1 | 17 | 3 | 36535.29 | 335 | 9.48 | 62.54 | 66.69 | 0.577 |
| FOXH1 | 12 | 2 | 51534.91 | 472 | 8.59 | 52.64 | 61.19 | 0.636 |
| FOXI1 | 12 | 2 | 42180.51 | 381 | 7.3 | 52.06 | 56.14 | 0.745 |
| FOXI2 | 13 | 2 | 41648.74 | 383 | 6.83 | 50.39 | 60.23 | 0.646 |
| FOXL1 | 18 | 1 | 40102.14 | 359 | 9.41 | 41.98 | 60.14 | 0.811 |
| FOXL2 | 15 | 2 | 32705.96 | 289 | 9.09 | 61.44 | 50.07 | 0.805 |
| FOXM1 | 4 | 9 | 69522.1 | 628 | 8.76 | 65.49 | 68.46 | 0.65 |
| FOXO1 | 15 | 4 | 69520.16 | 653 | 6.13 | 57.93 | 56.08 | −0.614 |
| FOXO3 | 20 | 4 | 68891.7 | 646 | 4.81 | 67.03 | 66.19 | 0.563 |
| FOXO4 | 14 | 3 | 65256.46 | 626 | 5.18 | 66.08 | 59.42 | 0.566 |
| FOXP1 | 6 | 25 | 73828.85 | 659 | 6.64 | 62.42 | 74.82 | 0.69 |
| FOXP2 | 4 | 21 | 67076.64 | 604 | 6.17 | 66.11 | 74.11 | 0.638 |
| FOXP3 | 8 | 13 | 50113.89 | 439 | 9.5 | 46.53 | 65.72 | 0.639 |

Based on the instability index, all proteins were deemed unstable, with values exceeding 40 (Table 2). Additionally, except for FOX O1, all FOX proteins were hydrophobic, reflected by their lower GRAVY values (Table 2). The genomic properties of FOX genes were also evaluated such as gene length (nt), chromosome length (bp), gene location on chromosome (start), gene location on chromosome (end), and the direction of strand.

## 2.4. Collinearity analysis of FOX genes in *L. rohita* and *H. sapiens*

The collinearity analysis revealed that the majority of FOX genes in *L. rohita* are distributed randomly across chromosomes 1, 2, 10, 18, and 23, while in *H. sapiens*, they are located on chromosomes 2, 7, 11, 14, 20, and Y. Most FOX genes are situated at the terminal ends of the chromosomes, with the exception of those on chromosomes 10 and 23 in *L. rohita* and on chromosomes 7, 11, and 20 in *H. sapiens* (Figure 3). The gray lines in the background represent collinear blocks within the genomes of *L. rohita* and *H. sapiens*, while the red lines highlight the syntenic FOX gene pairs.

## 2.5. Duplication and localization of FOX genes in *L. rohita*

To gain a clearer understanding of the evolutionary history of the FOX gene family in *L. rohita*, we investigated duplication events (Table 3). A total of seven duplicated gene pairs were identified, comprising segmental duplications (FOX I1/ FOX I2 and FOX D1/ FOX D3), tandem duplications (FOX A1/ FOX A2), and whole-genome duplications (FOX H1/ FOX M1, FOX F1/ FOX F2, FOX O1/ FOX O3, and FOX P1/ FOX P2) (Table 3). Additionally, these gene pairs underwent a Ka/ Ks ratio analysis to compare non-synonymous and synonymous divergence rates, which helps estimate the protein-coding potential of genomic regions. The Ka/ Ks ratio results showed that none of the values exceeded 1, suggesting that these proteins are under positive selection. Furthermore, all the genes were found to be influenced by purifying selection. The gene pairs with the longest estimated duplication times include FOX D1- FOX D3 (70.19 million years ago), FOX A1- FOX

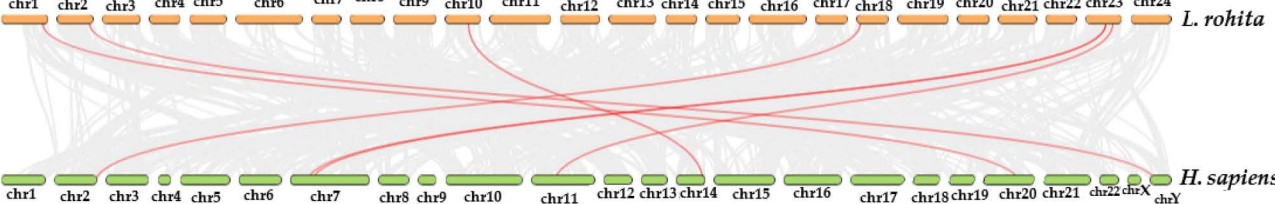

**Fig 3. Syntenic relationships between FOX genes in *L. rohita* and *H. sapiens*.** The gray lines in the background depict collinear blocks in the genomes of *L. rohita* and *H. sapiens*, while the red lines emphasize the syntenic pairs of FOX genes.

**Table 3. Ka/ Ks ratio analysis for each gene pair within the FOX gene family of *L. rohita*.**

| Gene Pairs | Chromosomes | Duplication | Ka | Ks | Ka/Ks | Selection | Time (MYA) |
|---|---|---|---|---|---|---|---|
| FOXA1-FOXA2 | 17/17 | TD | 0.3672 | 1.0966 | 0.334853183 | Purifying Selection | 49.85 |
| FOXI1-FOXI2 | 12/13 | SD | 0.2606 | 0.9325 | 0.279463807 | Purifying Selection | 42.39 |
| FOXD1-FOXD3 | 5/6 | SD | 1.1288 | 1.5442 | 0.730993395 | Purifying Selection | 70.19 |
| FOXH1-FOXM1 | 12/4 | WGD | 0.2762 | 0.6519 | 0.423684614 | Purifying Selection | 29.63 |
| FOXF1-FOXF2 | 18/2 | WGD | 0.2855 | 0.7335 | 0.389229721 | Purifying Selection | 33.34 |
| FOXO1-FOXO3 | 15/20 | WGD | 0.2691 | 0.6497 | 0.414191165 | Purifying Selection | 29.53 |
| FOXP1-FOXP2 | 6/4 | WGD | 0.272 | 0.9241 | 0.294340439 | Purifying Selection | 42.00 |

TD: Tandem Duplication; SD: Segmental duplication; WGD: Whole genome duplication; Ka: non-Synonymous substitutions; Ks: synonymous substitutions; MYA: Millions of years ago.

A2 (49.85 million years ago), FOX I1- FOX I2 (42.39 million years ago), and FOX P1- FOX P2 (42.00 million years ago), while more recent duplications include FOX F1- FOX F2 (33.34 million years ago), FOX M1- FOX H1 (29.63 million years ago), and FOX O1- FOX O3 (29.53 million years ago) (Table 3). Each gene is specifically located on chromosomes (Fig 4A), highlighting the relationships between FOX gene pairs that dictate the size, location, and orientation of connected genomic components. The localization and duplication events of FOX genes are illustrated in Figs 4A and 4B.

## 2.6. Prediction of Secondary and Tertiary Protein Structures

The prediction of secondary and three-dimensional structures was performed for all FOX proteins identified in the *L. rohita* genome. PSIPRED analysis revealed that the secondary structure of FOX proteins comprises random coils, beta twists, helices, and extended strands (S1 Fig). The secondary structural features were further assessed using the Phyre2 online tool, as presented in Table 4. The analysis indicated that the FOX gene structure has alpha helices constituting about 5–9%, beta sheets ranging from 0 to 6%. It was also revealed that the 2D structure of FOX proteins is also predominantly disordered (66–83%). Three-dimensional models for the FOX protein sequences were generated using Phyre2 (Fig 5) and subsequently validated with the SWISS MODEL online tool (S2 Fig). Both methods consistently confirmed the accuracy and robustness of our 3D structural predictions.

## 2.7. Scan PROSITE analysis

PROSITE was utilized to analyze FOX proteins for functional and structural residues associated with ProRule and the PROSITE signature. This tool helps identify intradomain features, such as active sites, binding sites, and disulfide bridges. The accuracy of functional predictions was enhanced by integrating the sensitivity of profiles with the specificity of motif recognition. Fig 6 illustrates the graphical representation of matches found in FOX proteins and the predicted features within those matches. The domain analysis of FOX D1, FOX G1, FOX P2, and FOX P3 proteins from the PROSITE database revealed various profiles alongside the Fork_head domain. In contrast, the FOX F2 protein displayed an additional range known as ZF-THAP. Additionally, FOX L1, FOX O1, and FOX O3 exhibited different profiles associated with the Fork_head domain. Among the other homologs in the FOX superfamily, only a single Fork_head domain was identified at varying amino acid residues.

## 2.8. Transcription factor binding sites (TFBSs) analysis

Transcriptional regulation within cells relies on transcription factors binding to specific genomic sites. The FOX gene superfamily in *L. rohita* was analyzed for its transcription factor binding sites (TFBSs) based on four transcription factors. Three promoter sites—GATA-1, AP-1, and p53—and one suppressor site, YY1, were selected for this study. Table 5 illustrates the distribution of TFBSs within the FOX gene superfamily in *L. rohita*. The analysis revealed a complex regulatory environment, with the following total numbers of binding sites for key transcription factors: AP-1 had 58, GATA-1 had 86, p53 had 85, and YY1 had 172 binding sites. These results highlight the ability of FOX genes to engage with a variety of transcription factors, indicating their important roles in gene regulation and cellular functions.

## 3. Discussion

In this study, we conducted a comprehensive analysis of the FOX gene family in *L. rohita*, focusing on their roles in development and stress response. To the best of our knowledge, this is the first systematic investigation of the FOX gene family in *L. rohita* that integrates phylogenetic analysis and computational genome characterization. Our findings provide significant insights into the diversity, structure, and functional significance of these genes. The identified differentially expressed FOX genes under various stress conditions offer valuable genetic resources for future studies on the adaptation and resilience of *L. rohita* to environmental challenges. Additionally, our multi-faceted approach, including the characterization of

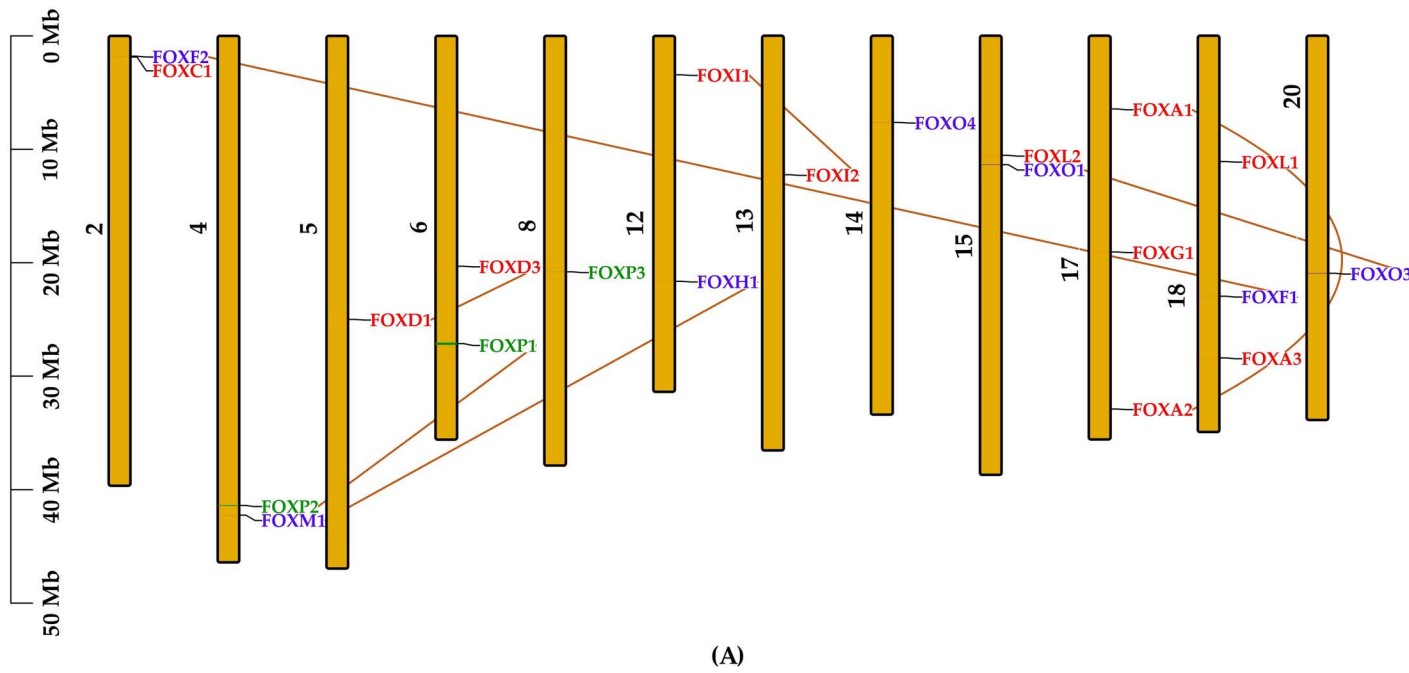

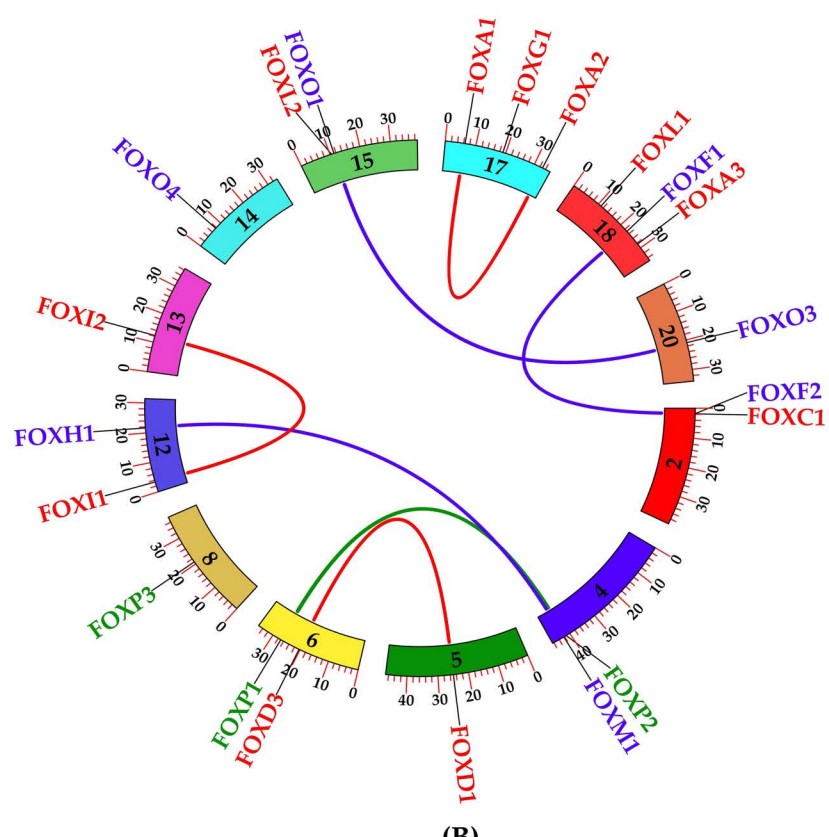

**Fig 4. Illustration of FOX superfamily gene localization and duplication events.** (A) The positions of FOX genes across all 12 chromosomes of *L. rohita* are shown, with mega-bases (Mb) used as the scale on the left. The corresponding chromosomal numbers are indicated at the center of each bar.

(B) A Circos plot depicts the FOX genes in the *L. rohita* genome, highlighting gene duplication events. Duplicated regions are shown in proximity due to the arrangement of the chromosomes, with segmentally duplicated genes connected by a direct communication pathway.

**Table 4. Predicted secondary structure characteristics of the FOX gene superfamily using Phyre2.**

| Sr. No | Name of Protein | Alpha Helix | Beta strand | Disorder |
|---|---|---|---|---|
| 1 | FOXA1 | 5% | 1% | 78% |
| 2 | FOXA2 | 5% | 1% | 80% |
| 3 | FOXA3 | 5% | 1% | 81% |
| 4 | FOXC1 | 5% | 1% | 83% |
| 5 | FOXD1 | 9% | 2% | 78% |
| 6 | FOXD3 | 8% | 1% | 79% |
| 7 | FOXF1 | 6% | 2% | 80% |
| 8 | FOXF2 | 9% | 6% | 66% |
| 9 | FOXG1 | 7% | 2% | 77% |
| 10 | FOXH1 | 6% | 2% | 82% |
| 11 | FOXI1 | 9% | 1% | 80% |
| 12 | FOXI2 | 9% | 0% | 81% |
| 13 | FOXL1 | 6% | 2% | 74% |
| 14 | FOXL2 | 9% | 2% | 74% |
| 15 | FOXM1 | 5% | 2% | 75% |
| 16 | FOXO1 | 6% | 2% | 82% |
| 17 | FOXO3 | 6% | 2% | 82% |
| 18 | FOXO4 | 7% | 1% | 82% |
| 19 | FOXP1 | 27% | 1% | 82% |
| 20 | FOXP2 | 28% | 1% | 83% |
| 21 | FOXP3 | 20% | 8% | 66% |

gene family members and their evolutionary dynamics, has broadened our understanding of the FOX genes' involvement in both developmental processes and stress responses across different physiological states of the species. These insights lay a foundation for further research on the molecular mechanisms driving development and stress tolerance in *L. rohita*.

The phylogenetic analysis of the FOX gene superfamily in *L. rohita* reveals valuable insights into the evolutionary trajectory of these genes. A maximum likelihood method was used to construct a phylogenetic tree based on 94 amino acid sequences from five species: *L. rohita*, *O. niloticus*, *C. idella*, *D. rerio*, and *H. sapiens*. This approach is consistent with previous research, which highlights the conservation of specific motifs within the FOX gene subclasses across species, pointing to a shared ancestral lineage for these genes [13]. The clustering of the FOX gene superfamily in *L. rohita* with *D. rerio* and *C. idella* suggests a common evolutionary origin, supporting earlier studies on the phylogenetic relationships among forkhead transcription factors across various taxa [14]. The identification of the 11 major clades—FOXH, FOXM, FOXP, FOXO, FOXF, FOXL, FOXG, FOXD, FOXC, FOXI, and FOXA—further underscores the conserved nature of this gene family, with previous investigations noting that specific FOX genes retain evolutionarily conserved functions across species, strengthening the idea of a unified evolutionary pathway [15]. The genetic similarity between *L. rohita* and other teleost species, like *D. rerio*, implies parallel evolution of FOX genes under similar selective pressures, aligning with findings on the conservation of gene regulatory elements across vertebrates [16]. Additionally, the high degree of sequence similarity between *L. rohita* and other teleosts reflects the evolutionary preservation of these genes, as demonstrated

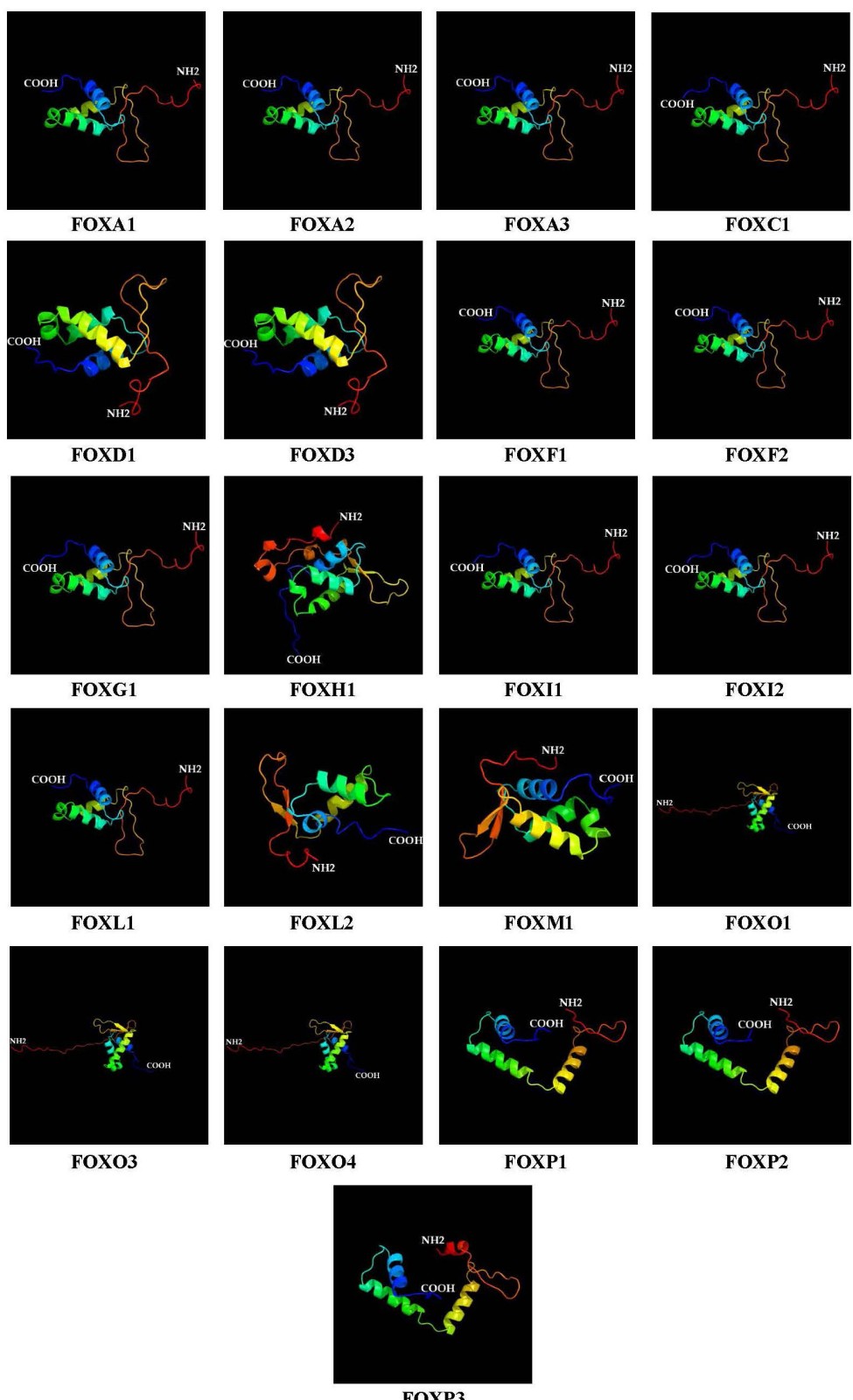

**Fig 5. Three-dimensional structure of the FOX gene superfamily in _L. rohita_ generated using Phyre2.**

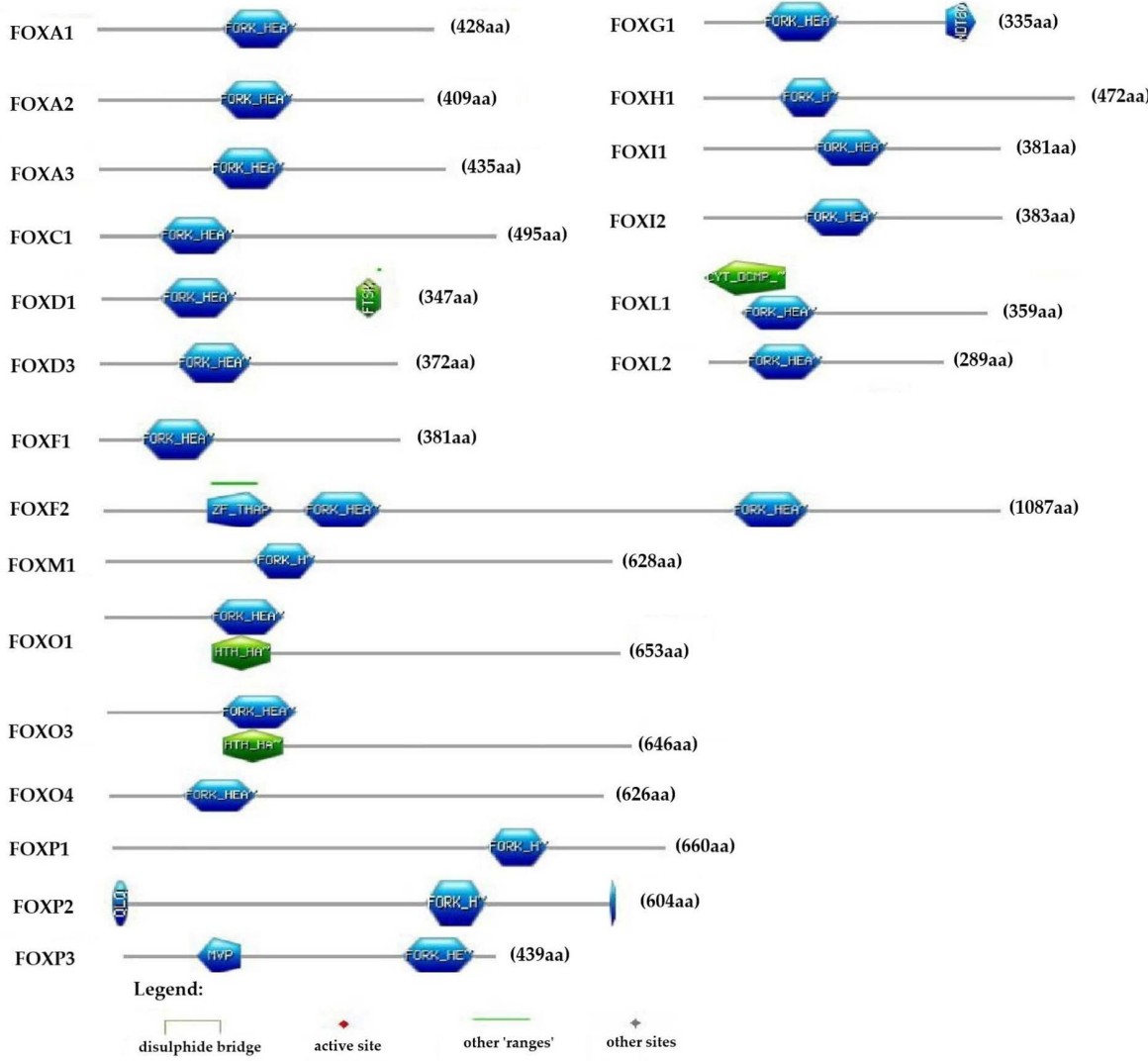

**Fig 6. Scan PROSITE identified intra-domain predictions.** This image represents all detected features and matches of the specified type. Pattern hits are displayed as thin colored bars without text, while hit profiles are represented as colored shapes accompanied by their corresponding PROSITE names.

in subsequent analyses of conserved motifs and gene structures [17]. Previous studies have explored the evolutionary dynamics of FOX genes, emphasizing their roles in development, and further highlight the importance of comparing functional roles across species. This comparative analysis is crucial for understanding the stress response and developmental processes in *L. rohita*, similar to the functions of FOX genes observed in other biological contexts [1].

The characterization of the FOX gene family in *L. rohita* has unveiled significant structural features with potential implications for gene function. Through motif analysis using MEME, ten conserved motifs were identified, with the most notable being MEME-6, which contains the Forkhead Box (FOX) protein domain. This domain plays a key role in DNA binding and transcriptional regulation, a feature that is well-established as characteristic of FOX family members due to their highly conserved DNA-binding forkhead domain, essential for various biological functions [18]. The conservation of the Forkhead Box domain across all FOX family members in *L. rohita* highlights its critical role in regulating processes such as cell

**Table 5. Transcription Factor Binding Sites (TFBSs) of FOX genes in *L. rohita*, highlighting the 4 TFBSs under study, total sites present, associated amino acid sequences and their localization in each gene.**

| Gene Name | TFBSs | Total Sites | Amino Acid Sequence | Amino Acid Localization |
|---|---|---|---|---|
| FOXA1 | AP-1 | 2 | TAAAGTCA, TGACTAAT | 520-527, 537-544 |
| | GATA-1 | 4 | GGAGATAA,TTATCTGA, TTATCAAA, TTATCGAT | 355-362, 436-443, 500-507, 778-785 |
| | p53 | 5 | GGGCTTT, GGGCCAT, TAGGCCC, GGGCATT, GGGCTGT | 260-266, 294-300, 930-936, 1383-1389, 1464-1470 |
| | YY1 | 7 | ATGGTT, AACCAT, ATGGGG, ATGGTC, GGCCAT, ATGGTG, TTCCAT | 54-59, 115-120, 256-261, 270-275, 295-300, 337-342, 549-554 |
| FOXA2 | AP-1 | 1 | GATAGTCA | 865-872 |
| | GATA-1 | 1 | TTATCTGG | 1408-1415 |
| | p53 | 2 | GGGCGCA, GCAGCCC | 670-676, 1151-1157 |
| | YY1 | 5 | TACCAT, ACCCAT, ATGGAT, ATGGAT, ATGGAT | 583-588, 631-636, 1198-1203, 1257-1262, 1269-1274 |
| FOXA3 | AP-1 | 1 | TGACTGAT | 937-944 |
| | GATA-1 | 2 | GACGATAA, TTATCCGT | 945-952, 1299-1306 |
| | p53 | 8 | GGGCTTT, CATGCCC, TAAGCCC, GGGCTTT, ACAGCCC, GGGCAAA, GGGCCTG, GGGCGGT | 1-7, 702-708, 873-879, 1187-1193, 1285-1291, 1398-1404, 1437-1443, 1446-1452 |
| | YY1 | 6 | ATGGCA, CTCCAT, AGCCAT, CGCCAT, TTCCAT, ATGGAC | 160-165, 256-261, 267-272, 1250-1255, 1311-1316, 1361-1366 |
| FOXC1 | AP-1 | – | – | – |
| | GATA-1 | 3 | TTATCATT, AATGATAA, GCAGATAA | 158-165, 479-486, 527-534 |
| | p53 | 10 | GGGCACA, CATGCCC, GGGCGGA, TAGGCCC, GGGCTTT, TCTGCCC, GGGCTTT, GGGCTGG, GGGCTGG, ACGGCCC | 25-31, 42-48, 242-248, 313-319, 712-718, 805-811, 820-826, 1035-1041, 1335-1341, 1397-1403 |
| | YY1 | 14 | GTCCAT, ATGGTC, ATGGCA, AACCAT, TGCCAT, ATGGCA, ATGGTG, ACCCAT, ATGGGC, TCCCAT, ATGGAC, AGCCAT, ATGGTT, TCCCAT | 86-91, 90-95, 524-529, 572-577, 605-610, 640-645, 667-672, 696-701, 818-823, 934-939, 978-983, 998-1003, 1293-1298, 1408-1413 |
| FOXD1 | AP-1 | 6 | TGACTATT, TGACTTAT, TGACTGAT, TGACTTTT, TCGAGTCA, TGACTGTT | 24-31, 844-851, 972-979, 993-1000, 1269-1276, 1353-1360 |
| | GATA-1 | 5 | AATGATAAT, TCTGATAAT, TTTATCATT, TTTATCATG, TTTATCTCT | 128-136, 194-202, 723-731, 1177-1185, 1416-1424 |
| | p53 | 5 | TAGGCCC, TTGGCCC, GGGCAAG, TGTGCCC, TCCGCCC | 54-60, 680-686, 913-919, 926-932, 1490-1496 |
| | YY1 | 6 | CTCCAT, ATGGTT, TTCCAT, ATGGCT, ATGGAA, ATGGGT | 432-437, 479-484, 1002-1007, 1013-1018, 1077-1082, 1183-1188 |
| FOXD3 | AP-1 | 1 | GCAAGTCA | 1015-1022 |
| | GATA-1 | 2 | AGAGATAAG, ATCGATAAG | 349-357, 527-535 |
| | p53 | 2 | GGGCAGA, GGGCGAG | 505-511, 1227-1233 |
| | YY1 | 7 | ATGGTA, ATGGTG, TCCCAT, ATGGCT, TTCCAT, ATGGTC, ATCCAT | 35-40, 444-449, 708-713, 837-842, 945-950, 1303-1308, 1358-1363 |
| FOXF1 | AP-1 | 5 | GAAAGTCA, TGACTAAA, GTCAGTCA, TGACTTCA, TGACTTTC | 187-194, 375-382, 613-620, 628-635, 826-833 |
| | GATA-1 | 4 | GCTGATAAA, ACTGATAAC, AGAGATAAA, ACAGATAAG | 3-11, 106-114, 342-350, 1480-1488 |
| | p53 | 4 | GGGCGCT, CTAGCCC, GGGCTTT, GGGCGGT | 18-24, 551-557, 983-989, 1420-1426 |
| | YY1 | 7 | AACCAT, GCCCAT, TTCCAT, ATGGGC, ATGGTG, GTCCAT, ATGGAA | 78-83, 554-559, 731-736, 981-986, 996-1001, 1425-1430, 1474-1479 |
| FOXF2 | AP-1 | 2 | TTTAGTCA, TGACTGCA | 259-266, 912-919 |
| | GATA-1 | 7 | ATTATCCCC, TTTATCTCC, TTTGATAAA, AATGATAAT, GTTATCCAT, CAAGATAAT, TTTATCCAG | 321-329, 498-506, 523-531, 762-770, 854-862, 941-949, 1153-1161 |

*(Continued)*

| Gene Name | TFBSs | Total Sites | Amino Acid Sequence | Amino Acid Localization |
|---|---|---|---|---|
| | p53 | 7 | GGGCTTT, GGGCGAT, GGGCTGA, ATTGCCC, ACAGCCC, TTCGCCC, TTCGCCC | 39-45, 454-460, 490-496, 576-582, 1163-1169, 1207-1213, 1450-1456 |
| | YY1 | 7 | ATCCAT, ATGGCC, ATGGTA, ATGGAA, CCCCAT, CGCCAT, CGCCAT | 857-862, 996-1001, 1064-1069, 1118-1123,1216-1221, 1339-1344, 1421-1426 |
| FOXG1 | AP-1 | 2 | TTCAGTCA, TGACTGGG | 343-350, 1492-1499 |
| | GATA-1 | 6 | TTTATCTGA, CTTATCTAA, GTTATCTCT, TTTAT-CACG, CTTATCAAA, ATTATCAAA | 175-183, 562-570, 617-625, 801-809, 912-920, 1411-1419 |
| | p53 | 2 | CAGGCCC, GGGCCTT | 214-220, 558-564 |
| | YY1 | 6 | TTCCAT, CTCCAT, ATGGTG, ATGGTA, ATGGAT, AGCCAT | 395-400, 646-651, 955-960, 1154-1159, 1384-1389, 1483-1488 |
| FOXH1 | AP-1 | 6 | AAGAGTCA, ACTAGTCA, TCGAGTCA,ACTAGTCA, TGACTGTA, TGACTTTT | 42-49, 55-62, 571-578, 983-990, 1036-1043,1293-1300 |
| | GATA-1 | 4 | TTTGATAAC, TTTATCACA, ATTATCACA, GTTATCATT | 123-131, 608-616, 616-624, 1475-1483 |
| | p53 | – | – | – |
| | YY1 | 5 | CTCCAT, ATGGTA, ATGGCT, TTCCAT, TGCCAT | 467-472, 648-653, 1089-1094, 1110-1115, 1313-1318 |
| FOXI1 | AP-1 | 4 | TTGAGTCA, TTGAGTCA, AAAAGTCA, TGTAGTCA | 653-660, 665-672, 682-689, 1115-1122 |
| | GATA-1 | 4 | AAAGATAAA, GATGATAAT, ATTGATAAA, CTTATCGCA | 13-21, 501-509, 544-552, 1303-1311 |
| | p53 | 5 | ACTGCCC, GGGCTAT, CATGCCC, GCGGCCC, GGGCACT | 356-362, 631-637, 1020-1026, 1413-1419, 1455-1461 |
| | YY1 | 3 | AACCAT, ATGGGC, ATCCAT | 392-397, 629-634, 1246-1251 |
| FOXI2 | AP-1 | 3 | TGACTTAA, TGACTGAT, TGACTTTA | 40-47, 299-306, 1350-1357 |
| | GATA-1 | 3 | TTTATCCTC, TTTATCGTA, TCGGATAAT | 6-14, 56-64, 859-867 |
| | p53 | 1 | TTAGCCC | 1293-1299 |
| | YY1 | 9 | ATCCAT, ATGGAA, ATGGTT, ATGGGA, ATGGCT, AACCAT, TGCCAT, TACCAT, ATGGTT | 96-101, 259-264, 275-280, 524-529, 728-733, 852-857, 895-900, 936-941, 1164-1169 |
| FOXL1 | AP-1 | 1 | TAGAGTCA | 525-532 |
| | GATA-1 | 4 | TTTATCCAG, ATCGATAAG, ATTGATAAA, GTGGATAAG | 248-256, 399-407, 418-426, 1090-1098 |
| | p53 | 1 | GGTGCCC | 1468-1474 |
| | YY1 | 14 | TTCCAT,ACCCAT, ATGGAT, TACCAT, ATGGAT, ATGGAT, ATGGCA, ATGGGT, ATGGTA, ATGGAA, TACCAT,ATGGTT, ACCCAT, ATGGAT | 212-217, 438-443, 452-457, 490-495, 589-594, 754-759, 792-797,831-836, 908-913, 928-933, 958-963, 998-1003, 1067-1072, 1257-1262 |
| FOXL2 | AP-1 | 2 | GGCAGTCA, TGACTCTA | 400-407, 732-739 |
| | GATA-1 | 9 | TGTGATAAT, ACTGATAAG, TTTGATAAT, CTAGATAAT, TTTATCATG, TAAGATAAT, ATTATCATA, ATTATCCAC, AGAGATAAC | 225-233, 239-247, 516-524, 737-745, 755-763, 995-1003, 1040-1048, 1202-1210, 1454-1462 |
| | p53 | 7 | ACTGCCC, AATGCCC, GGGCAAT, GGGCCTA, GGGCAGT, TTAGCCC, GGGCTGA | 150-156, 171-177, 195-201, 368-374, 773-779, 1224-1230, 1316-1322 |
| | YY1 | 11 | ATGGTG, CACCAT, GACCAT, ATGGAA, ATGGTT, ATCCAT, ATGGGA, ATGGAA, ATGGGT, ATGGCC, ATGGAC | 82-87, 334-339, 476-481, 625-630, 640-645,690-695, 761-766, 809-814, 831-836, 899-904, 1079-1084 |
| FOXM1 | AP-1 | 3 | TGACTGTC, TGACTATT, ACCAGTCA | 63-70, 654-661, 1230-1237 |
| | GATA-1 | 3 | TGGGATAAA, TTTATCGTA, GTGGATAAA | 418-426, 728-736, 794-802 |
| | p53 | 8 | GGGCGCC, GGGCATT, GGGCGGG, GGGCTTG, ATGGCCC, ATCGCCC, TCCGCCC, GGGCTAG | 473-479, 530-536, 1021-1027, 1026-1032, 1033-1039, 1358-1364,1425-1431, 1448-1454 |

*(Continued)*

| Gene Name | TFBSs | Total Sites | Amino Acid Sequence | Amino Acid Localization |
|---|---|---|---|---|
| | YY1 | 18 | AGCCAT, AGCCAT, ATGGTG, ATGGCA, GTCCAT, GTCCAT, ATGGGC, TGCCAT, ATGGTG, ATGGCG, GACCAT, CCCCAT, ATGGAA, ATGGCC, ATGGGT, CACCAT, GCCCAT, TGCCAT | 154-159, 215-220, 376-381, 426-431, 445-450, 463-468, 471-476, 614-619, 777-782, 807-812, 886-891, 939-944, 1006-1011, 1033-1038, 1067-1072, 1389-1394, 1428-1433, 1441-1446 |
| FOXO1 | AP-1 | 5 | TGACTCAA, CGTAGTCA, TGACTTCG, TGACTATT, TGCAGTCA | 680-687, 827-834, 1236-1243, 1245-1252, 1330-1337 |
| | GATA-1 | 1 | GTCGATAAA | 1315-1323 |
| | p53 | 6 | ATTGCCC, ATAGCCC, CCCGCCC, GGGCAAA, GGGCTTC, GGGCGGT | 74-80, 88-94, 371-377, 577-583, 607-613, 1473-1479 |
| | YY1 | 6 | TACCAT,ATGGTT,GACCAT, ATGGCA, ATGGTC, AGCCAT | 36-41, 40-45, 243-248, 776-781, 870-875, 1325-1330 |
| FOXO3 | AP-1 | 3 | ACCAGTCA, TGACTGGT, TGACTTGT | 443-450, 814-821, 1392-1399 |
| | GATA-1 | 7 | ATTATCATC, ATCGATAAT, CAAGATAAA, CTTATCAAG, CCAGATAAT, CCAGATAAT, TCCGATAAA | 300-308, 764-772, 994-1002, 1171-1179, 1282-1290, 1295-1303, 1357-1365 |
| | p53 | 4 | GGGCTCT, GGGCGCA, GTGGCCC, GGGCAGG | 1373-1379, 1435-1441, 1442-1448, 1481-1487 |
| | YY1 | 9 | AGCCAT, ATGGGT, ATGGTT, ATGGGT, ATGGTT, ATGGAA, ATCCAT, ATGGGC, ATGGGC | 29-34, 108-113, 331-336, 452-457, 523-528,715-720, 1030-1035, 1371-1376, 1433-1438 |
| FOXO4 | AP-1 | 6 | TGACTGAT, TGACTTTC, TGACTGTA, TGACTTTA, TGACTGAC, TGACTGCA | 132-139, 795-802, 1158-1165, 1167-1174, 1474-1481, 1478-1485 |
| | GATA-1 | 6 | CTTATCAAA, TTTATCCAT, CGTGATAAA, TCAGATAAC, TCAGATAAG, TCAGATAAA | 17-25, 61-69, 635-643, 762-770, 1027-1035, 1137-1145 |
| | p53 | 3 | GGGCAAG, ATCGCCC, CTAGCCC | 161-167, 459-465, 477-483 |
| | YY1 | 13 | ATCCAT,ATGGTT,ATGGTA, CACCAT, ATGGGG, TTCCAT, ATGGTT,ATGGGG, CCCCAT, ACCCAT, GACCAT, ACCCAT, TCCCAT | 64-69, 99-104, 178-183, 224-229, 405-410, 414-419, 427-432, 556-561, 627-632, 1122-1127, 1245-1250, 1464-1469, 1488-1493 |
| FOXP1 | AP-1 | 1 | GACAGTCA | 549-556 |
| | GATA-1 | – | – | – |
| | p53 | 3 | GGGCTCC, TGGGCCC, GGGCCCG | 103-109, 486-492, 487-493 |
| | YY1 | 4 | AACCAT, AACCAT, TTCCAT, ATGGGT | 349-354, 605-610, 917-922, 1438-1443 |
| FOXP2 | AP-1 | 1 | CCAAGTCA | 388-395 |
| | GATA-1 | 6 | TGTGATAAA, GTTATCAGA, ATTATCAAT, AAAGATAAA, CGTGATAAA, CTCGATAAA | 200-208, 348-356, 619-627, 989-997, 1221-1229, 1341-1349 |
| | p53 | – | – | – |
| | YY1 | 5 | ATCCAT,GACCAT, ATCCAT, ATGGGG, ATGGCA | 165-170, 487-492, 491-496, 875-880, 1250-1255 |
| FOXP3 | AP-1 | 3 | TGACTCAG, AGAAGTCA, TAGAGTCA | 93-100, 319-326, 363-370 |
| | GATA-1 | 5 | GTTATCACC, TTGGATAAA, CACGATAAA, TTTATCCAG, AGTGATAAA | 223-231, 473-481, 729-737, 1112-1120, 1343-1351 |
| | p53 | 2 | GGGCATT, TTTGCCC | 385-391, 1101-1107 |
| | YY1 | 10 | TACCAT,ACCCAT, CTCCAT, ATGGTT, CTCCAT, ATGGCT, AGCCAT, GCCCAT, ATGGTC, TGCCAT | 15-20, 73-78, 148-153, 372-377, 398-403, 590-595, 996-1001, 1104-1109, 1337-1342, 1404-1409 |

differentiation and metabolism [19]. This conserved motif suggests that FOX genes in *L. rohita* likely perform roles similar to those seen in other vertebrates. Prior studies have confirmed that FOX family genes are involved in diverse biological processes across species, reinforcing the idea of a conserved functional role [20].

Gene structure analysis further revealed notable variations in exon-intron arrangements among the FOX genes in *L. rohita*, as well as differences in the untranslated regions (UTRs) upstream and downstream of coding sequences. These variations point to potential functional diversity within the FOX genes, possibly influenced by the unique environmental pressures faced by *L. rohita* in its aquatic habitat [19]. The presence of multiple exons and introns suggests opportunities for alternative splicing, a mechanism critical for enhancing the functional complexity of genes. This idea aligns with previous research that highlights the involvement of FOX proteins in alternative splicing regulation through interactions with spliceosomal components [21]. Furthermore, variations in UTRs may contribute to the differential regulation of gene expression, with earlier studies showing that intron presence can influence tissue-specific splicing and gene expression patterns [22]. Understanding these structural elements is essential for predicting how FOX genes may respond to environmental cues, such as stressors in aquaculture environments, where differential regulation could be crucial for organismal adaptation [19].

The physicochemical analysis of FOX proteins in *L. rohita* revealed a broad range of molecular weights, from 32705.96 to 119782.93 Da, reflecting the diversity typical of FOX gene family proteins. This range is consistent with findings from previous studies on teleost species, which also highlight the evolutionary diversity of these proteins in terms of molecular size [23]. The pI values of these proteins, ranging from 4.81 to 9.50, suggest that *L. rohita* FOX proteins possess varying charge properties, potentially influencing their interactions with DNA and other cellular components. The importance of charge variability in protein interactions, particularly within aquatic species, has been well-documented [24].

Most of the FOX proteins exhibited aliphatic index values above 41.83, indicating a high degree of thermostability, which is crucial for maintaining functionality across diverse environmental conditions. This attribute is particularly beneficial in aquaculture, where stressors like temperature fluctuations and hypoxia can challenge protein function. The role of thermostability in enhancing protein resilience to thermal stress is supported by earlier research showing a correlation between high aliphatic index values and increased protein stability under stress [25]. Despite this, instability index values for the majority of FOX proteins exceeded 40, suggesting that these proteins may experience rapid turnover or conformational changes in response to environmental signals—a common characteristic of transcription factors involved in stress response and development [26].

Additionally, the analysis showed that most FOX proteins in *L. rohita* are hydrophobic, as indicated by their low GRAVY (Grand Average of Hydropathy) values, implying their potential involvement in membrane-associated activities or protein-protein interactions requiring hydrophobic interfaces. This aligns with previous research, which emphasizes the critical role of hydrophobic interactions in the functionality of membrane-associated proteins in aquatic organisms [27]. Understanding these physicochemical properties is key to predicting how FOX proteins might respond to environmental stressors in aquaculture environments, where rapid adaptation is essential.

The duplication and localization analysis of FOX genes in *L. rohita* has shed light on the evolutionary development of this gene family. Seven duplicated gene pairs were identified in this study, including segmental, tandem, and whole-genome duplications, which contribute to the expansion and functional diversification of gene families, allowing them to play specialized roles in various physiological processes. The Ka/Ks ratio analysis indicated that these duplicated genes are under purifying selection, suggesting that they have preserved essential functions throughout evolutionary time. Previous studies have similarly shown that duplicated genes often retain their original functions or develop complementary roles over time, supporting the notion of functional conservation [28].

Gene localization analysis revealed that FOX genes in *L. rohita* are distributed across various chromosomes, with a notable clustering of genes at the terminal ends of chromosomes. This distribution is consistent with patterns observed in other species, such as *H. sapiens*, where syntenic gene pairs are conserved across species. This suggests that FOX

genes may have similar functional roles across different organisms, an idea supported by research on FOX gene conservation and synteny [1]. Additionally, conserved FOX gene clusters have been observed in bilaterians, reinforcing the concept of evolutionary conservation and functional similarity among these genes across various species [29].

The evolutionary history of the FOX gene family in *L. rohita* appears to have been influenced by species-specific chromosomal rearrangements, likely driven by serial gene duplications. This observation aligns with earlier findings that FOX gene clusters have evolved through such mechanisms, highlighting the dynamic nature of gene family evolution [15]. The insights gained from this study provide a valuable framework for future research on the functional roles of FOX genes in *L. rohita*, particularly in relation to environmental stressors and other physiological challenges.

The transcription factor binding site (TFBS) analysis of FOX genes in *L. rohita* uncovered a complex regulatory framework. We identified binding sites for several key transcription factors, including AP-1, GATA-1, p53, and YY1. Notably, the analysis revealed 58 binding sites for AP-1, 86 for GATA-1, 85 for p53, and 172 for YY1, indicating that FOX genes may regulate a broad spectrum of cellular processes, such as cell cycle regulation, apoptosis, and stress response. This is consistent with earlier studies showing the role of AP-1 in modulating gene expression in response to diverse stimuli, including stress [30]. Similarly, GATA-1 is widely recognized for its involvement in hematopoiesis and stress regulation [31]. The abundance of binding sites for these transcription factors underscores the critical role of FOX genes in mediating gene expression, particularly under stress conditions.

This regulatory complexity suggests that FOX genes may serve as integrators of multiple signaling pathways, enabling responses to environmental stressors like oxidative stress, hypoxia, and temperature changes. The capacity of FOX genes to engage with several transcription factors hints at their pivotal role in stress-response mechanisms. Previous research also highlights how transcription factors like p53 and YY1 influence cellular stress responses through cis- and trans-regulatory elements, coordinating responses to changing environmental conditions [32]. The involvement of p53, a well-established tumor suppressor that governs cell cycle regulation and apoptosis, further emphasizes the importance of FOX genes in maintaining cellular integrity under stress [33]. YY1, functioning as both a transcriptional repressor and activator, adds another layer of regulatory complexity, particularly through histone modifications essential for stress adaptation [33]. Understanding these regulatory interactions is crucial for deciphering the role of FOX genes in *L. rohita*, especially in relation to stress-response pathways. The findings indicate that FOX genes contribute not only to fundamental cellular processes but also to the organism's adaptive responses to fluctuating environmental conditions. This intricate regulatory network highlights the evolutionary significance of FOX genes in teleosts, as they navigate diverse and often challenging habitats.

The strong correlation between phylogenetic clustering, conserved motifs, gene duplication, and transcriptional regulation highlights the evolutionary pressures shaping FOX genes in *L. rohita*, optimizing their roles in development, stress adaptation, and cellular homeostasis. The identification of conserved motifs within duplicated gene clusters suggests that these expansions have not led to functional redundancy but rather contributed to regulatory diversification, allowing fine-tuned gene expression in response to environmental fluctuations. This is further reinforced by the presence of multiple stress-responsive transcription factor binding sites, such as AP-1, p53, and GATA-1, which are crucial for oxidative stress regulation, immune response, and developmental processes. Additionally, the physicochemical properties of FOX proteins, including their broad molecular weight range, thermostability, and predominantly hydrophilic nature, align with their involvement in signal transduction, transcriptional regulation, and protein-protein interactions necessary for maintaining cellular function under stress. The clustering of FOX genes in chromosomal regions known to facilitate regulatory interactions further underscores their evolutionary significance. Collectively, these findings suggest that FOX genes in *L. rohita* have maintained structural and functional integrity through strong evolutionary conservation while simultaneously acquiring regulatory flexibility via gene duplication and transcription factor interactions. This interplay between conservation and adaptability likely provides *L. rohita* with enhanced resilience to environmental challenges, reinforcing the critical role of FOX genes in both developmental stability and adaptive potential.

While this study provides valuable insights, the limitations of this research should be acknowledged. The in-silico analysis of the FOX gene superfamily in *L. rohita* offers important perspectives into their genomic landscape, evolutionary significance, and roles in development and stress response. However, several limitations must be considered. Potential inaccuracies in gene annotation could impact the identification and characterization of these genes, leading to incomplete or misleading findings. Furthermore, the absence of experimental validation for the predicted gene functions restricts a deeper understanding of the biological relevance and specific roles of these FOX genes. Without laboratory-based confirmation, the predictions remain theoretical. Additionally, this study does not consider environmental influences, which may have a significant effect on gene expression and regulation under natural conditions. Finally, technical limitations inherent in bioinformatics tools, such as algorithmic biases, may introduce errors into the analysis.

To fill these gaps, future research on the FOX gene family in *L. rohita* should focus on exploring the functional roles of these genes in various biological processes, with a particular emphasis on their response to environmental stressors such as temperature fluctuations, hypoxia, and pollution. To validate these roles, we could use cutting-edge techniques like CRISPR-Cas9 or RNA interference to either knock down or overexpress specific genes. This would provide clearer evidence of how FOX genes help the fish adapt to stress and support normal development. It would also be helpful to explore how FOX genes interact with other important signaling pathways, such as those involved in the body's response to oxidative stress or immune challenges. This could give us a more complete picture of how FOX genes maintain balance within cells. To dig deeper, we could look at transcriptomic and proteomic data under various stress conditions, which would help us identify specific targets regulated by FOX genes. Moreover, it's worth considering the potential of using FOX gene variants in selective breeding programs to create stronger, more resilient fish strains. This could help improve both the productivity and stress resistance of *L. rohita* and other fish species. Lastly, conducting comparative genomics across different fish species would provide valuable insights into how the FOX gene family has evolved, shedding light on its role in helping these species adapt to different environments.

## 4. Materials and Methods

This study has been approved by the ethical committee on animal rights and welfare, Department of Zoology, Government Sadiq College Women University, Bahawalpur 63100, Punjab, Pakistan (No. 12/DZ/GSCWUB/2023)

### 4.1. Identification and Characterization of FOX Genes in the *L. rohita* Genome

To obtain the complete genome sequence files for the study species *L. rohita*, *O. niloticus*, *C. idella*, *D. rerio*, and *H. sapiens*, the National Center for Biotechnology Information (NCBI) Genome database (https://www.ncbi.nlm.nih.gov/) was consulted. Human sequences served as the query to retrieve non-redundant genomic, proteomic, and coding sequence data related to the *L. rohita* FOX gene superfamily, with the corresponding identification numbers (IDs) listed in S1 Table. This retrieval was performed using the Basic Local Alignment Search Tool (BLAST) and Hidden Markov Model (HMM) searches, applying an E value threshold of ≤1.0 × e-5 [34]. Additionally, the Forkhead box protein domain was identified through the InterPro online tool (https://www.ebi.ac.uk/interpro/) by utilizing various supplementary programs [35]. After gathering the relevant protein sequences, duplicates were eliminated to avoid confusion [36]. The resulting non-redundant sequences were analyzed with the Simple Modular Architecture Research Tool (SMART) (http://smart.embl-heidelberg.de/), and the NCBI Conserved Domain Database (CDD) (https://www.ncbi.nlm.nih.gov/Structure/cdd/cdd.shtml) was employed to search for conserved FOX protein domains in *L. rohita* proteins [37].

The physicochemical properties, such as isoelectric point (pI), grand average of hydropathy (GRAVY), molecular weight (MW), aliphatic index (AI), and instability index (II), were determined using Expasy ProtParam tool (https://web.expasy.org/protparam/) [38]. Chromosome numbers were identified through genomic sequence data using the NCBI database [39].

## 4.2. Phylogenetic Analysis of FOX genes superfamily

The 94 amino acid sequences were aligned using the MUSCLE algorithm in MEGA11. A phylogenetic tree was constructed using the maximum likelihood method with the Jones-Taylor-Thornton (JTT) model and 1,000 bootstrap replicates to ensure robustness. The resulting Newick file was exported and visualized using the online tool iTOL (https://itol.embl.de/login.cgi) [40]. This analysis elucidated the evolutionary relationships among *L. rohita* and the reference species *O. niloticus*, *C. idella*, *D. rerio*, and *H. sapiens* based on their physical and genetic similarities and differences.

## 4.3. Syntax gene duplication and chromosomal distribution of FOX superfamily

The occurrence of duplications in the FOX gene family of *L. rohita* was assessed through pairwise alignment of the FOX genes, utilizing the MUSCLE algorithms available in MEGA11 v.11.0. [41]. Additionally, the pairwise synonymous substitutions per synonymous site (Ks) and non-synonymous substitutions per non-synonymous site (Ka) [42], adjusted for multiple hits, were calculated using the DnaSP v10.0 software (http://www.ub.edu/dnasp/) [43]. Furthermore, the age of all duplicated genes was estimated using the formula $t = Ks/2r$ where r is $1.1 \times 10^{-8}$ [44]. The FOX genes of both *L. rohita* and *H. sapiens* were also aligned to create a dual synteny plot [45]. Genomic resources and the General Feature Format (GFF) file [46] were utilized to identify the chromosomal gene locations within the *L. rohita* genome using the TBtools program (https://github.com/CJ-Chen/TBtools) as a display server. Chromosomal and genomic data served as input for plotting Circos using the same TBtools program [47]. Circos illustrated the variations in genomic organization and the overall positional relationships among the genomic intervals of *L. rohita* FOX genes [48].

## 4.4. Characterization of structural features

The conserved motifs were analyzed using the MEME Suite (https://meme-suite.org/meme/), with all *L. rohita* FOX protein sequences submitted as queries in FASTA format. The MEME Suite was also employed to identify 10 conserved motif patterns within the FOX gene [49]. A hit data file obtained from the Conserved Domain online tool (https://www.ncbi.nlm.nih.gov/Structure/cdd/wrpsb.cgi) was used for domain analysis in TBtools. The GTF file facilitated gene structure visualization within TBtools. Ultimately, TBtools produced a comprehensive graph that integrated these three analyses with a phylogenetic tree, offering a holistic view of the conserved motifs, exon-intron structures, and functional domains of the FOX genes [50,51].

## 4.5. Secondary structure and 3D models of FOX superfamily proteins

We began by predicting protein secondary structures using the PSIPRED online server (http://bioinf.cs.ucl.ac.uk/psipred/) renowned for its accuracy [52,53]. These 2D predictions were further validated with Pyre2 (https://www.sbg.bio.ic.ac.uk/phyre2/) [54], which confirmed the observed structural elements. For three-dimensional modeling, we employed Phyre2, a robust web-based homology modeling platform [54,55]. To ensure the consistency and accuracy of our 3D model, we subsequently validated it using the Swiss-Model online tool (https://swissmodel.expasy.org/interactive) [56].

## 4.6. Scan Prosite analysis

The online tool Scan Prosite (https://prosite.expasy.org/scanprosite/) was employed to identify functional and structural intra-domains. The protein sequence was submitted to query the Prosite motif collection for analysis [57].

## 4.7. Retrieval and identification of transcription factor binding sites (TFBSs)

Genomic files containing only the promoter sequences of 1500 bp for each gene were submitted to the PROMO software (https://alggen.lsi.upc.es/cgi-bin/promov3/), which utilizes version 6.4 of TRANSFAC to identify potential binding sites in the sequences [58]. This analysis focused on three transcriptional promoter binding sites: GATA-1, AP-1, and p53, as well

as one repressor site, YY1. The GATA-1 transcription factor is crucial for development, as it interacts with FOX genes to regulate gene expression during the development and functioning of erythroid cells (red blood cells) and other hematopoietic lineages [59,60]. AP-1 is responsible for regulating gene expression in response to various stimuli, including stress. It can modulate the activity of FOXO and FOXA proteins in reaction to oxidative stress, promoting either cell survival or apoptosis [61,62]. The p53 protein is activated under stress conditions, such as DNA damage or oxidative stress, and it can regulate FOXO1 and FOXO3, thereby enhancing their roles in apoptosis and cell cycle arrest [63]. YY1 is known to repress the expression of FOX O genes, particularly during stress responses like oxidative stress [64]. Additionally, YY1 can regulate developmental processes by inhibiting genes that are involved in cell differentiation [65].

## 5. Conclusions

To sum up, this study presents a comprehensive computational analysis of the FOX gene family in *L. rohita*, shedding light on their crucial roles in development and response to stress. By exploring the diversity, structure, and functions of these genes, we have uncovered important insights into how they have evolved, driven by genetic variation, duplication, and selective forces. This comprehensive approach, combining phylogenetic analysis with genome characterization, highlights the FOX genes' key role in helping *L. rohita* adapt to environmental challenges. The discovery of these genes not only deepens our understanding of their evolutionary importance but also provides valuable genetic resources for genetic enhancement and sustainable management of *L. rohita*. These findings create a solid foundation for future work in selective breeding and aquaculture, while also paving the way for deeper exploration into the molecular mechanisms that govern the species' development, stress response, and resilience.

## Supporting information

**S1 Table. Genomic properties of the FOX gene superfamily in *L. rohita*.**
(DOCX)

**S1 Fig.**
(PDF)

**S2 Fig.**
(PDF)

## Acknowledgments

The authors are thankful to the Ongoing Research Funding (ORF-2025-833), King Saud University, Riyadh, Saudi Arabia

## Author contributions

**Conceptualization:** Saima Naz.

**Data curation:** Saima Naz, Marco Ragni.

**Formal analysis:** Marco Ragni.

**Investigation:** Urwah Ishaque, Ahmad Manan Mustafa Chatha.

**Methodology:** Urwah Ishaque, Ahmad Manan Mustafa Chatha, Marco Ragni.

**Resources:** Ibrahim Alhidary.

**Software:** Shabana Naz.

**Validation:** Qudrat Ullah.

**Visualization:** Qudrat Ullah.

**Writing – original draft:** Shabana Naz, Muhammad Farooq, Ibrahim Alhidary.

**Writing – review & editing:** Shabana Naz, Muhammad Farooq, Ibrahim Alhidary.

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
