## [Decision Letter · Decision Letter 0]

11 Feb 2025

Dear Dr. Naz,

Thank you for submitting your manuscript to PLOS ONE. After careful consideration, we feel that it has merit but does not fully meet PLOS ONE’s publication criteria as it currently stands. Therefore, we invite you to submit a revised version of the manuscript that addresses the points raised during the review process.

We look forward to receiving your revised manuscript.

Kind regards,

Bijay Kumar Behera, Ph.D.

Academic Editor

PLOS ONE

Journal Requirements:

Additional Editor Comments:

The recent manuscript has its own merit. According to the two reviewers recommendations, my decision is to Major revision of the manuscript. Kindly see the two reviewers comment.

Reviewers' comments:

Reviewer's Responses to Questions

**Comments to the Author**

1. Is the manuscript technically sound, and do the data support the conclusions?

Reviewer #1: Yes

Reviewer #2: Yes

2. Has the statistical analysis been performed appropriately and rigorously?

Reviewer #1: Yes

Reviewer #2: Yes

3. Have the authors made all data underlying the findings in their manuscript fully available?

Reviewer #1: Yes

Reviewer #2: Yes

4. Is the manuscript presented in an intelligible fashion and written in standard English?

Reviewer #1: Yes

Reviewer #2: No

Reviewer #1: The article “Investigating the Role of FOX Gene Family in Development and Stress Response in Labeo rohita: A Multi-faceted Analysis of Phylogeny and Genome Characterization” is well written.

The authors need to consider a few points mentioned below before publication.

•The Abstract section is not justified.

•L499: If possible, for secondary structure, authors are advised to use more than one program for their validation with suitable references

•For using iTol, the file type used for designing the phylogeny should be mentioned.

•The study only highlights the available sequences of the FOX gene in the NCBI database. The authors haven’t confirmed the 5’-UTR and 3’-Poly A tail by using RACE PCR. Using the terms in conclusion, like detailed genome-wide analysis, might not be true. If possible, avoid using terms like that.

• The discussion section is well written. The authors advised to revise accordingly.

Reviewer #2: Comments about the manuscript:

The manuscript entitled “Investigating the role of FOX gene family in development and stress response in Labeo rohita: a multi-faceted analysis of phylogeny and genome characterization” aims to explore the evolutionary and molecular roles of FOX genes in L. rohita using computational analysis. The introduction is streamlined and clear but the other section should be improved. The manuscript may be published with a revision as detailed below.

•In Abstract, authors should specifically state the need of this research.

•Line No 134-135: Bootstrap consensus values were calculated for each node in the analysis……………. In Figure 1, no bootstrap values are visible.

•Figure caption should be improved with details.

•Discussion section should be revised by the authors. Try to correlate the results obtained with each other.

•Line No 499: Authors are advised to use multiple programs for validation with appropriate references.

•4.2. Phylogenetic Analysis of FOX genes superfamily: Authors should write their methodology in details.

•Line No 528: Avoid the term ‘detailed genome-wide analysis of the FOX gene family’, because authors only used computational tools for this study.

**Do you want your identity to be public for this peer review?** For information about this choice, including consent withdrawal, please see our Privacy Policy

Reviewer #1: No

Reviewer #2: No

---

## [Author Response · Author response to Decision Letter 1]

27 Mar 2025

Dear Editor and Reviewers,

We would like to express our sincere gratitude for the time and effort invested in reviewing our manuscript entitled “Investigating the Role of FOX Gene Family in Development and Stress Response in Labeo rohita: A Multi-faceted Analysis of Phylogeny and Genome Characterization.” Your insightful comments have been invaluable in enhancing the quality and clarity of our work. Below, we provide a detailed, point-by-point response to all comments raised.

Response to Reviewer #1

Comment: “The Abstract section is not justified.”

Response: We have revised the abstract to ensure full justification in accordance with the journal’s formatting guidelines. The updated abstract now meets the required presentation style.

Comment: “If possible, for secondary structure, authors are advised to use more than one program for their validation with suitable references.”

Response: We thank the reviewer for the suggestion. Our manuscript now specifies that FOX protein secondary structure was predicted using PSIPRED and independently validated with Pyre2, with both methods yielding consistent results. All relevant references are now provided in the manuscript.

Comment: “For using iTol, the file type used for designing the phylogeny should be mentioned.”

Response: We have now specified in the Methods section that the phylogenetic tree was generated using the Newick format file generated by MEGA11.

Comment: “The study only highlights the available sequences of the FOX gene in the NCBI database. The authors haven’t confirmed the 5’-UTR and 3’-Poly A tail by using RACE PCR. Using the terms in conclusion, like detailed genome-wide analysis, might not be true. If possible, avoid using terms like that.”

Response: We acknowledge this concern and have revised the manuscript accordingly. The text now clearly states that the study is based on computational analysis of sequences available in the NCBI database. The term “detailed genome-wide analysis” has been replaced with “comprehensive computational analysis” to more accurately reflect the scope of our work.

Comment: “The discussion section is well written. The authors advised to revise accordingly.”

Response: We appreciate the positive feedback. Minor revisions have been made in the discussion to enhance clarity and to improve the integration and correlation of the results.

Response to Reviewer #2

Comment: “In Abstract, authors should specifically state the need of this research.”

Response: The abstract has been updated to explicitly state the motivation behind our study, highlighting the significance of understanding the role of FOX genes in the developmental and stress response mechanisms of Labeo rohita.

Comment: “Bootstrap consensus values were calculated for each node in the analysis……………. In Figure 1, no bootstrap values are visible.”

Response: We have revised Figure 1 to include bootstrap values for each node. The figure legend has been updated to describe these values and the method of calculation in detail.

Comment: “Figure caption should be improved with details.”

Response: The caption for Figure 1 has been expanded to include additional details.

Comment: “Discussion section should be revised by the authors. Try to correlate the results obtained with each other.”

Response: We have restructured the discussion to provide a more cohesive interpretation of our findings. The revised version more clearly correlates the outcomes.

Comment: “Authors are advised to use multiple programs for validation with appropriate references.”

Response: In our study, FOX protein secondary structure was accurately predicted using PSIPRED and validated with Pyre2 for 2D analysis. We have now incorporated SWISS-MODEL for 3D structure validation, and relevant references have been added to the manuscript.

Comment: “4.2. Phylogenetic Analysis of FOX genes superfamily: Authors should write their methodology in details.”

Response: We have expanded Section 4.2 to include a detailed description of our phylogenetic analysis methodology to enhance reproducibility.

Comment: “Avoid the term ‘detailed genome-wide analysis of the FOX gene family’, because authors only used computational tools for this study.”

Response: In line with the feedback, we have replaced the term “detailed genome-wide analysis” with “comprehensive computational analysis” to more accurately represent our study’s scope.

We believe that these revisions have significantly improved the manuscript and have addressed all the concerns raised. We appreciate the reviewers’ constructive feedback and are hopeful that the revised manuscript meets the journal’s standards for publication.

Thank you for considering our revised manuscript. We look forward to your positive response.

---

## [Decision Letter · Decision Letter 1]

8 Apr 2025

Dear Dr. Naz,

Thank you for submitting your manuscript to PLOS ONE. After careful consideration, we feel that it has merit but does not fully meet PLOS ONE’s publication criteria as it currently stands. Therefore, we invite you to submit a revised version of the manuscript that addresses the points raised during the review process.

We look forward to receiving your revised manuscript.

Kind regards,

Bijay Kumar Behera, Ph.D.

Academic Editor

PLOS ONE

Journal Requirements:

Additional Editor Comments:

Dear Author,

Now the manuscript has been revised. One minor comment is in the manuscript. Kindly see the reviewers comments and revise accordingly.

Reviewers' comments:

Reviewer's Responses to Questions

**Comments to the Author**

Reviewer #1: All comments have been addressed

Reviewer #2: All comments have been addressed

2. Is the manuscript technically sound, and do the data support the conclusions?

Reviewer #1: Yes

Reviewer #2: Yes

3. Has the statistical analysis been performed appropriately and rigorously?

Reviewer #1: Yes

Reviewer #2: Yes

4. Have the authors made all data underlying the findings in their manuscript fully available?

Reviewer #1: Yes

Reviewer #2: Yes

5. Is the manuscript presented in an intelligible fashion and written in standard English?

Reviewer #1: Yes

Reviewer #2: Yes

Reviewer #1: (No Response)

Reviewer #2: The authors have addressed all the comments satisfactorily. Only one issue I have found in the Revised manuscript which is mentioned below:

Line no 137-138: These sequences were organized into 12 major clades ……….. But there are 11 clades. In Figure 1, they have also clustered the phylogenetic tree into 11 clades. Please rectify it.

The manuscript may be accepted after fixing the issue.

**Do you want your identity to be public for this peer review?** For information about this choice, including consent withdrawal, please see our Privacy Policy

Reviewer #1: No

Reviewer #2: No

---

## [Author Response · Author response to Decision Letter 2]

11 Apr 2025

Dear Editor and Reviewers,

We would like to express our sincere gratitude for the time and effort you have dedicated to reviewing our manuscript entitled “Investigating the Role of FOX Gene Family in Development and Stress Response in Labeo rohita: A Multi-faceted Analysis of Phylogeny and Genome Characterization.” Your thoughtful and constructive feedback has been instrumental in improving the overall quality and clarity of our work.

Following the recent round of minor revision, we have carefully addressed the remaining comment provided by Reviewer #2, as detailed below:

Response to Reviewer #2

Comment:

Line no. 137–138: “These sequences were organized into 12 major clades...” But there are 11 clades. In Figure 1, they have also clustered the phylogenetic tree into 11 clades. Please rectify it.

Response:

The mentioned line has been corrected in the revised manuscript, changing the number of clades from 12 to 11 to accurately reflect the phylogenetic analysis and Figure 1. We have reviewed the manuscript thoroughly to ensure consistency throughout the text.

We appreciate your continued support and are hopeful that the manuscript is now suitable for acceptance. Thank you once again for your guidance and constructive suggestions throughout the review process.

Sincerely,

Shabana Naz

Department of Zoology, Government Sadiq College Women University, Bahawalpur 63100, Punjab, Pakistan.

Email: drshabananaz@gcuf.edu.pk

---

## [Editor Report · Decision Letter 2]

15 Apr 2025

Investigating the Role of FOX Gene Family in Development and Stress Response in Labeo rohita: A Multi-faceted Analysis of Phylogeny and Genome Characterization

PONE-D-24-60377R2

Dear Dr. Naz,

We’re pleased to inform you that your manuscript has been judged scientifically suitable for publication and will be formally accepted for publication once it meets all outstanding technical requirements.

Kind regards,

Bijay Kumar Behera, Ph.D.

Academic Editor

PLOS ONE

Additional Editor Comments (optional):

The manuscript entitled "Investigating the Role of FOX Gene Family in Development and Stress Response in Labeo rohita: A Multi-faceted Analysis of Phylogeny and Genome Characterization" has been accepted for publication.
---

## [Editor Report · Acceptance letter]

PONE-D-24-60377R2

PLOS ONE

Dear Dr. Naz,

I'm pleased to inform you that your manuscript has been deemed suitable for publication in PLOS ONE. Congratulations! Your manuscript is now being handed over to our production team.

Kind regards,

on behalf of

Dr. Bijay Kumar Behera

Academic Editor

PLOS ONE